# Integrative QTL Mapping and Transcriptomic Profiling to Identify Growth-Associated QTL and Candidate Genes in Hong Kong Catfish (*Clarias fuscus*)

**DOI:** 10.3390/ani15121707

**Published:** 2025-06-09

**Authors:** Yian Zhu, Dayan Zhou, Yijun Shen, Chuanhao Pan, Yu Chen, Yulei Zhang, Binlan Yang, Guangli Li, Huapu Chen, Changxu Tian

**Affiliations:** 1Guangdong Research Center on Reproductive Control and Breeding Technology of Indigenous Valuable Fish Species, Guangdong Provincial Engineering Laboratory for Mariculture Organism Breeding, Guangdong Provincial Key Laboratory of Aquatic Animal Disease Control and Healthy Culture, Fisheries College, Guangdong Ocean University, Zhanjiang 524088, China; chuyian@126.com (Y.Z.); syjhaida@163.com (Y.S.); panch@gdou.edu.cn (C.P.); chenyu200003@126.com (Y.C.); zhangyl@gdou.edu.cn (Y.Z.); ligl@gdou.edu.cn (G.L.); chenhp@gdou.edu.cn (H.C.); 2Guangxi Introduction and Breeding Center of Aquaculture, Nanning 530001, China; magiczdyan@126.com (D.Z.); 13768884250@163.com (B.Y.)

**Keywords:** *Clarias fuscus*, growth traits, transcriptome, QTL mapping, candidate gene

## Abstract

The Hong Kong catfish (*Clarias fuscus*) is a freshwater species commonly cultured in southern China for its high nutritional value. Despite its economic importance, research on the genes associated with growth traits in *C. fuscus* is limited, hindering the development of effective breeding strategies. In this study, quantitative trait loci (QTL) mapping and RNA-seq were performed to investigate the growth traits of *C. fuscus*. By combining these two approaches, 27 growth-related candidate genes were identified within 17 QTL intervals, which play important roles in the growth activities of fish. The identification of these QTL regions and candidate genes not only fills the gap in research on growth-related genes in *C. fuscus*, but also provides valuable data to further elucidate the genetic mechanisms underlying growth traits in this species.

## 1. Introduction

Growth traits are among the most critical economic features in aquaculture species. Enterprises focusing on fish with superior growth performance can benefit from reduced rearing cycles, lower production costs, and significant increases in yield and economic returns. Consequently, selecting and breeding fish species with exceptional growth potential has become a central goal for the sustainable development of the aquaculture industry [1]. Growth traits are regulated by numerous small-effect polygenes, known as quantitative trait loci (QTL). While most of these loci have minor effects, a few key genes exert a substantial influence on growth [2]. Several genes involved in growth regulation have been identified in fish. Genes within the growth hormone/insulin-like growth factor (GH/IGF) axis, such as growth hormone (*gh*), growth hormone receptor (*ghr*), insulin-like growth factor 1 (*igf-I*), insulin-like growth factor 2 (*igf-II*), and insulin-like growth factor binding proteins (*igfbps*), play critical roles in regulating metabolic processes and cell growth [3]. Additionally, members of the myogenic regulatory factors (MRFs) gene family, including myogenin (*myog*), myogenic differentiation 1 (*myod*), and myogenic factor 5 (*myf-5*), are primarily expressed in vertebrate skeletal muscle and are crucial for muscle development through the regulation of cell proliferation and the activation of muscle-specific genes [4]. Other genes, such as melanocortin receptor-4 (*mc4r*), influence growth by regulating energy metabolism, while gastric hunger hormone (*ghrelin*) plays a role in growth regulation by enhancing feeding and digestive efficiency [5,6]. These genes interact through a complex signaling network that collectively influences growth-related phenotypic variation in fish. Given the species-specific nature of growth regulation, screening growth-related genes across different species is essential for gaining a comprehensive understanding of the genetic mechanisms behind growth regulation in fish.

QTL mapping is a crucial tool for investigating the genetic basis of economic traits in aquaculture species, playing a key role in guiding the identification of candidate genes [7]. Traditional breeding methods rely on the careful selection of individuals with desirable characteristics followed by successive generations of breeding to produce seedlings with superior economic traits. However, this process is time-consuming, costly, and highly susceptible to external factors such as environmental influences, limiting its practical application in production [8]. In contrast, QTL mapping enables the identification of complex traits by pinpointing specific genomic regions, facilitating the early selection of superior individuals. This approach not only significantly shortens the breeding cycle but also improves breeding efficiency and enhances economic returns. Currently, QTL mapping is widely applied in breeding studies of aquaculture species, addressing key economic traits such as sex determination [9,10], hypoxia tolerance [11,12], and body color [13,14]. For growth traits, QTL mapping studies have been conducted in various aquaculture species, including common carp (*Cyprinus carpio*) [15] and mirror carp (*Cyprinus carpiovar. specularis*) [16]. These studies have enriched the genetic resources for fish breeding and provided valuable insights into the genetic basis of growth traits in fish.

While QTL mapping is valuable for identifying chromosomal regions linked to growth traits by analyzing the genetic structure of economic traits, it primarily locates broader chromosomal regions where candidate genes are mapped. Without further fine-mapping, its ability to provide detailed candidate gene information is limited [17]. On the other hand, RNA-seq offers insights into gene expression dynamics, revealing how gene expression varies across individuals of different sizes [18,19]. In recent years, transcriptomics analysis has been widely used to identify the differentially expressed genes (DEGs) associated with growth traits in aquaculture species, for example, rainbow trout (*Oncorhynchus mykiss*) [20], mandarin fish (*Siniperca chuatsi*) [21], and fine flounder (*Paralichthys adspersus*) [22]. Combining QTL mapping with RNA-seq allows for the narrowing of the candidate gene range through co-mapping strategies like eQTL analysis, leading to more accurate gene mapping. For instance, a co-analysis in giant grouper (*Epinephelus Lanceolatus*) identified 34 candidate genes related to growth and six growth-related QTL [23]. Similarly, in Pacific whiteleg shrimp (*Litopenaeus vannamei*), four candidate genes were identified from nine growth-related QTLs using conjoint analysis [24]. This integrated approach improves the identification of growth-related candidate genes and enhances our understanding of the genetic mechanisms underlying growth traits in aquaculture species.

The Hong Kong catfish (*Clarias fuscus*) is the only native species of the Clariidae family in China [25]. This species is highly valued for its desirable traits, including strong adaptability, tender meat, and a high content of essential amino acids with a lowly fat content [26]. It has become one of the most important economic aquaculture species in southern China [27]. However, the development of the aquaculture industry for this species is hindered by the lack of targeted selective breeding programs. Therefore, there is an urgent need to enhance key economic traits, particularly growth, and to select seedlings with superior growth potential. Currently, the whole-genome sequences and high-density genetic linkage maps of *C. fuscus* have been constructed [25,26,28]. The acquisition of these whole-genome sequences and high-density genetic maps has laid the groundwork for identifying economic traits in this species. However, the genes related to the growth traits remain poorly understood. In this study, we performed QTL mapping for eight growth traits of *C. fuscus* using the established genetic map [28] and integrated muscle RNA-seq data to identify growth-related candidate genes. The findings will provide a scientific foundation for understanding the genetic basis of growth traits in *C. fuscus* and contribute to the development of *C. fuscus* lines with improved growth performance.

## 2. Materials and Methods

### 2.1. Ethics Statement

All experimental protocols in this study were conducted in accordance with the standards of the Guangdong Ocean University Animal Research and Ethics Committee (NIH Pub. No.85-23, revised in 1996). The subjects of this study were not endangered or protected species.

### 2.2. Sample Preparation and Phenotyping

To efficiently identify the genomic regions associated with target quantitative traits, genetic correlations within a full-sib F1 family of *C. fuscus* were utilized for QTL mapping. The family used for the QTL mapping of growth traits in this study originated from the same group previously employed for constructing the genetic linkage map [28]. This mapping family of *C. fuscus* was provided by the Guangxi Hongtai Aquatic Product Farm, Guangxi Province, China. A full-sib F1 family was generated via artificial fertilization with one dam and one sire. The F1 family was cultured in ponds and fed pellet feed three times daily. From this family, 200 F1 individuals were randomly selected for genetic linkage map construction [28], and these individuals were subsequently used for the QTL mapping of growth traits. In this study, eight growth-related traits were measured: body weight (BW), body height (BH), body length (BL), body width (BWI), orbital diameter (OD), caudal peduncle length (CPL), caudal peduncle height (CPH), and pre-dorsal length (PDL). The raw data for these traits were preprocessed using Microsoft Excel, followed by statistical analysis, including descriptive statistics, correlation analysis, and analysis of variance (ANOVA), conducted using SPSS 26.0.

For transcriptomic analysis, a mixed family of *C. fuscus* was selected. This mixed family, also provided by the Guangxi Hongtai Aquatic Product Farm, was reared at the freshwater aquaculture facility of Guangdong Ocean University. One-month-old fish of a similar size were transferred to the facility and cultured in a single pond under identical conditions, with pellet feed provided three times daily and water changed three times a week. At 6 months of age, 254 individuals were randomly selected for body weight measurement. Based on weight, nine individuals with the highest and nine with the lowest values were chosen to form three large-sized groups (B1, B2, B3) and three small-sized groups (S1, S2, S3). The average body weight of the large-sized groups was 100.13 ± 12.96 g, while that of the small-sized groups was 32.29 ± 7.48 g. The fish were fasted for 24 h before sample collection. Before dissection, the fish were anesthetized with eugenol (Shandong Keyuan Biochemical Co., Ltd., Heze, China). Muscle tissues were harvested, rapidly frozen in liquid nitrogen, and stored at −80 °C. For each group, muscle tissues from three individuals were mixed to a single sample for subsequent sequencing and analysis.

### 2.3. QTL Mapping for Growth Traits

QTL mapping for growth traits was conducted using the genetic linkage map of *C. fuscus* [28]. Eight growth traits were analyzed using the Permutation Test (PT) in MapQTL 6.0. The PT was performed with 1000 permutations to determine the LOD threshold for each trait at a 95% confidence level. The results from the PT were combined with the actual QTL significance levels obtained from the analysis to establish uniform thresholds for the eight growth traits. Interval mapping (IM) in MapQTL 6.0 was then used to analyze these traits, and the corresponding QTL segments were identified based on the established thresholds [29].

### 2.4. Screening of Candidate Genes for Growth Traits QTL and Functional Enrichment Analysis

Candidate genes and functional annotations within the QTL intervals for growth traits were identified using the reference genome of *C. fuscus* (https://www.ncbi.nlm.nih.gov/datasets/genome/GCA_030347435.1/, accessed on 23 June 2024). Functional genes within the QTL intervals identified through linkage analysis were selected based on this reference genome. The corresponding sequences within each QTL region were extracted and annotated by BLASTN (v2.7.1 +) analysis. Subsequently, Gene Ontology (GO) and Kyoto Encyclopedia of Genes and Genomes (KEGG) enrichment analyses were performed on the annotated genes using TBtools software (v2.210). The resulting visualizations were generated using the R package clusterProfiler (v3.8.1) [30].

### 2.5. Transcriptome Library Creation and Raw Data Processing

Total RNA was extracted from muscle tissues using TRIzol^®^ Reagent (Invitrogen, Carlsbad, CA, USA) following the manufacturer’ s protocol. Genomic DNA was removed using RNase-Free DNase I (Thermo Fisher Scientific, Waltham, MA, USA). RNA concentration and purity were assessed using a NanoDrop ΙΙΙ 2000 spectrophotometer (Thermo Fisher Scientific, Waltham, MA, USA), and RNA integrity was confirmed using an Agilent 2100 Bioanalyzer (Agilent Technologies, Santa Clara, CA, USA). Only RNA samples with RIN > 7.0 were used for library construction. mRNA was randomly fragmented using NEB Fragmentation Buffer (New England Biolabs Co., Ltd, Beijing, China). The first strand of cDNA was synthesized using random oligonucleotides as primers in the M-MuLV reverse transcriptase system (New England Biolabs Co., Ltd, Beijing, China). The RNA strand was then degraded with RNaseH, and the second strand of cDNA was synthesized with dNTPs in the DNA polymerase I system. The purified double-stranded cDNA underwent end repair, A-tail addition, and ligation of sequencing junctions. cDNAs of approximately 250–300 bp were selected using AMPure XP beads (Shanghai Via-geneprobio Technologies Co., Ltd., Shanghai, China), followed by PCR amplification, and the PCR products were purified again. The libraries were then constructed [30]. Real-time PCR was performed for quality testing on the purified products. After passing the quality control, sequencing was conducted on an Illumina NovaSeq 6000 platform (Illumina, San Diego, CA, USA), generating 150 bp paired-end reads. The sequencing method was based on the ‘Sequencing by Synthesis’ principle. Raw sequencing data were processed using fastp (version 0.23.4) to filter out low-quality reads, reads containing adapters, and reads with poly-N sequences, resulting in clean reads for further analysis. [31]. These high-quality reads were aligned to the reference genome using HISAT2 software (v2.0.5), ensuring fast and accurate alignment to obtain the localization information of the reads on the reference genome (https://www.ncbi.nlm.nih.gov/datasets/genome/GCA_046453815.1/, accessed on 2 February 2025) [32]. Finally, new transcripts were assembled using StringTie software (v1.3.3b) [33].

### 2.6. Screening of DEGs and Functional Enrichment Analysis

The big-sized-individual group was used as the experimental group, while the small-sized-individual group served as the control group. Differential expression analysis between the two groups was conducted using the DESeq2 R package (version 1.6.3), identifying differentially expressed genes (DEGs) with a *p*-value < 0.05 and |fold change| ≥ 1. Gene Ontology (GO) and Kyoto Encyclopedia of Genes and Genomes (KEGG) enrichment analyses of the DEGs were performed using TBtools software (v2.210), and the results were visualized using the R package clusterProfiler (v3.8.1).

### 2.7. QTL and Transcriptome Association Analysis

By integrating the results from the QTL mapping analysis of significant regions and transcriptome differential gene analysis, a set of common genes between the two datasets was identified for further quantitative analysis and expression profiling. For gene expression analysis, expression levels were log2-transformed, and heatmaps were generated using the R package pheatmap (v3.5.0). The heatmaps were constructed using a fully interlocked clustering method and the Euclidean distance measurement metric.

### 2.8. Real-Time Fluorescence Quantitative PCR (qRT-PCR) Validation

The expression of candidate genes in the muscle of *C. fuscus* was quantified using qRT-PCR. The RNA used for transcriptome sequencing was reverse transcribed into complementary DNA (cDNA) using HiScript III All-in-One RT SuperMix (Vazyme Biotech Co., Ltd, Nanjing, China), optimized for quantitative PCR (qPCR). qRT-PCR was performed using PerfectStart Green qPCR SuperMix (Code#TG-AQ601-02) on a LightCycler^®^ 480 detection system (Roche, Basel, Switzerland), with each sample analyzed in triplicate. Primers for qPCR were designed using the Primer Premier 6.0, with the β-actin gene serving as the internal reference gene. The reaction mixture was prepared at a total volume of 10 μL, consisting of 5.0 μL of 2× PerfectStart Green qPCR SuperMix, 1.0 μL of cDNA, 3.4 μL ddH2O, 0.3 μL of the forward primer, and 0.3 μL of the reverse primer. The reaction procedure was as follows: (1) Preincubation: 94 °C for 300 s; (2) amplification: (95 °C for 20 s, 60 °C for 20 s, 72 °C for 20 s) × 40 cycles; (3) melting: 95 °C for 10 s, 65 °C for 60 s, 97 °C for 1 s; and (4) cooling: 37 °C for 300 s. The comparative CT (2^−ΔΔCT^) method was used to assess relative gene expression levels. The primers for candidate genes and the internal control gene (β-actin) are shown in Appendix A.

## 3. Results

### 3.1. Morphological Statistics

In this study, eight morphological traits (BW, BL, BH, BWI, OD, CPL, CPH, and PDL) were measured in 200 F1 individuals of *C. fuscus*. The morphological analyses showed that the frequency distribution of these traits conformed to a normal distribution (Appendix A). The coefficients of variation (CV) for the eight traits ranged from 8.00% to 11.90% (Appendix A), indicating substantial variability, which made these traits suitable candidates for QTL analysis.

### 3.2. QTL Analysis and Candidate Gene Identification

Using QTL composite interval mapping with the MapQTL 6.0 software, we analyzed eight growth traits of *C. fuscus*. The analysis identified 17 QTLs (LOD > 3.5) distributed across eight linkage groups, including LG06, LG07, LG09, LG13, LG14, LG17, LG18, and LG20 (Figure 1). The peak positions of these QTLs ranged from 5.62 cM (qCPH-8) to 186.52 cM (qCPH-5). The phenotypic variance explained (PVE) by these QTLs ranged from 8.00% (qBL-4) to 11.90% (qCPH-6), with an average PVE of 9.04% (Table 1). Multiple growth trait QTLs were clustered on LG06, LG09, and LG17. Specifically, three QTLs for CPH, CPL, and BH were identified on LG06. On LG09, four QTLs were identified for BW, BL, PDL, and BWI, with a notable overlap between the significant regions for BW (41.931–45.835 cM) and BWI. Similarly, QTLs for BW, BWI, and PDL were on LG17, with the significant region for BW (13.520–17.098 cM) overlapping with that for BWI. A total of 162 candidate genes were identified within these 17 QTL intervals (Appendix A).

### 3.3. Functional Annotation of QTL Candidate Genes

To explore the functions and associated pathways of the QTL candidate genes, GO and KEGG enrichment analyses were performed. The GO analysis revealed five significantly enriched terms within the biological process (BP) category, including cell adhesion (GO:0007155) and RNA metabolic process (GO:0016070). In the molecular function (MF) category, three terms were significantly enriched, such as calcium ion binding (GO:0005509) and structural molecule activity (GO:0005198). No significant enrichment was observed in the cellular component (CC) category (Figure 2A). KEGG pathway analysis identified nine significantly enriched pathways, including Brite Hierarchies, Propanoate metabolism, and Transcription machinery (Figure 2B).

### 3.4. Quality Control of Transcriptome Sequencing Data

Following quality control procedures, low-quality reads from the raw sequencing data were removed, resulting in an average of 50,513,416 (98.25%) clean reads for the big-sized group and 44,737,262 (95.63%) clean reads for the small-sized group. The average Q20 and Q30 values for the big-sized individuals were 98.69% and 96.20%, respectively, while for the small-sized individuals, these values were 98.10% and 95.03%, respectively. After quality control, the clean reads from both groups were mapped to the reference genome of the *C. fuscus*. On average, 90.47% of the clean reads from the big-sized group and 83.57% from the small-sized group were successfully mapped to the reference genome (Table 2). These results indicate that the sequencing results are of high quality and suitable for subsequent analyses.

### 3.5. Differential Gene Expression Analysis Related to Body Size

Gene expression changes in muscle tissue samples from big-sized and small-sized individuals were compared, identifying a total of 3824 differentially expressed genes (DEGs) associated with individual size in *C. fuscus*. Of these, 2252 genes were upregulated, and 1572 genes were downregulated (*p* < 0.05 and |log_2_(fold change) | ≥ 1, Figure 3A). Table 3 presents the top 20 genes ranked by fold change between the big-sized- and small-sized-individual groups.

### 3.6. Functional Classification of DEGs

To further explore the biological significance of the DEGs, GO and KEGG enrichment analyses were conducted. The GO enrichment analysis revealed the significant enrichment of DEGs in two functional categories, with 94 subcategories identified within the biological process (BP) category and 16 subcategories within the cellular component (CC) category. No significant enrichment was observed in the MF category. The DEGs were significantly enriched in terms such as Envelope (GO:0031975), Mitochondrion (GO:0005739), ATP metabolic process (GO:0046034), and Translation (GO:0006412) (Figure 3B). KEGG pathway analysis indicated that DEGs were significantly enriched in nine pathways, including Carbon Metabolism, 2-Oxocarboxylic Acid Metabolism, and Glycolysis/Gluconeogenesis. Among these pathways, the Biosynthesis of Amino Acids pathway was the most significantly enriched, while the Insulin Signaling pathway had the highest number of enriched DEGs (Figure 3C).

### 3.7. Carbohydrate Metabolism Pathway Analysis

To investigate the molecular mechanisms underlying size variation in *C. fuscus*, we examined the expression patterns of key DEGs in major carbohydrate metabolism pathways across the two groups (Figure 4). The analysis revealed that genes involved in glycolysis and the tricarboxylic acid (TCA) cycle, such as glucose-6-phosphate isomerase (*gpi*), ATP-dependent 6-phosphofructokinase (*pfkm*), and fructose-1,6-bisphosphatase isozyme 2 (*fbp2*), were upregulated in the large-sized individuals. Conversely, genes associated with odd-chain fatty acid catabolism, isoleucine catabolism, and ethanol metabolism, including alcohol dehydrogenase class-3 (*adh3*) and peroxisomal acyl-coenzyme A oxidase 1 (*acox1*), exhibited higher expression levels in the small-sized individuals.

### 3.8. Integration of QTL Mapping and Transcriptome Data

By integrating the results of QTL mapping and transcriptome differential expression analysis, we identified 27 candidate genes associated with growth, including epidermal growth factor receptor (*egfr*), ataxin-7protein 3 (*atxn7*), glutamine-rich protein 1 (*qrich1*), and nucleophosmin (*npm1*) (Table 4). The expression analysis of the 27 growth candidate genes is shown in Figure 5A. To validate the expression patterns of these growth-related candidate genes, we performed qRT-PCR analysis on five randomly selected genes in muscle tissues from both big-sized and small-sized *C. fuscus* individuals. The expression patterns observed through qRT-PCR were consistent with those from the transcriptome data, confirming the accuracy and reliability of the transcriptome results (Figure 5B).

## 4. Discussion

Growth traits are critical economic traits in aquaculture species and have garnered considerable attention [34]. While existing studies have offered valuable insights into the regulatory mechanisms of growth genes in fish, the complexity of growth regulation has limited the identification of related genes. Combining transcriptome analysis with QTL mapping offers a promising approach for identifying genes associated with growth traits. Conjoint analyses have already been conducted on the growth traits of various fish species. For instance, in black porgy (*Acanthopagrus schlegelii*), 42 candidate genes for growth traits were identified through QTL mapping, and two key genes, *Magi1* and *Tp53inp2*, were further pinpointed via transcriptome analysis [35]. Similarly, transcriptome analysis of the muscles and liver of turbot (*Scophthalmus maximus*) identified 174 growth-related genes, which were subsequently narrowed down to 45 through joint QTL analysis [36]. These studies demonstrate that, unlike standalone QTL mapping, combining it with transcriptome analysis to identify DEGs allows for a more precise selection of candidate genes related to superior growth traits in fish. In this study, we identified QTLs associated with growth in *C. fuscus* by integrating transcriptome data related to body size, and pinpointed candidate genes and relevant pathways. This work provides a foundation for further investigation into the growth regulatory mechanisms in *C. fuscus* and offers key data for developing fast-growing strains.

### 4.1. QTL Mapping of Growth Traits in C. fuscus

Numerous studies have demonstrated that QTL mapping is a primary method for identifying quantitative trait loci and understanding the underlying mechanisms of quantitative traits [37]. One critical metric in QTL mapping is the proportion of variance explained (PVE), which quantifies the extent to which a QTL contributes to phenotypic trait variation. A higher PVE value indicates a greater influence of the QTL on the trait, while a lower PVE suggests a lesser effect, implying that other genetic or environmental factors may also significantly shape the trait [38]. In fish, growth traits are classic quantitative traits, with associated QTLs typically exhibiting relatively low PVE values. For example, mirror carp (9.6–24.2%) [16], crucian carp (*Carassius auratus*) (PVE, 10.1–13.2%) [39], ussuri catfish (*Pseudobagrus ussuriensis*) (11.9–20.5%) [40], and dusky kob (*Argyrosomus japonicus*) (9.3–29.5%) [41] all demonstrate this pattern.

Based on the available genetic map of *C. fuscus*, this study analyzed eight growth traits using QTL mapping. Except for OD, CPL, and BWI, all other traits were mapped to two or more significant QTL intervals, resulting in the identification of 17 growth-related QTLs. The PVE of these 17 QTLs ranged from 8.00 to 11.90%. These results suggest that growth traits in *C. fuscus* are complex and influenced by multiple genes. Compared to the major QTLs identified in other fish species, the individual QTLs identified in this study have a relatively smaller impact on growth traits. This could be due to the sample size used in this experiment. Previous studies have observed that as the sample size increases, the PVE values of QTLs tend to decrease [2]. This trend also seems to apply to the growth traits of fish. For instance, in crucian carp and black carp (*Mylopharyngodon piceus*), QTL mapping was performed using 102 and 128 individuals, respectively, yielding average PVE values of 11.30% and 12.56% [1,39]. In contrast, for studies of red-tail catfish (*Hemibagrus wyckioides*) and giant grouper, which used 167 and 216 individuals, respectively, the average PVE values were 9.43% and 7.28% [23,42]. This phenomenon may explain the generally lower PVE values observed for QTL in this study. Additionally, the relatively low PVE values may suggest that the growth traits of *C. fuscus* are influenced by a larger number of small-effect genes. Although the additive effects of a single gene are modest, the cumulative impact of multiple small-effect genes, along with potential non-additive effects such as dominance and epistasis, may significantly affect the phenotype [43,44,45]. Future studies could focus on quantifying the additive and non-additive effects of various QTLs, providing theoretical insights and practical guidance for marker-assisted selection (MAS). Moreover, the QTL intervals for a single growth trait in *C. fuscus* are more concentrated compared to other fish species. For example, two BW-related QTLs in this study were mapped to LG09 and LG17, while in red-tail catfish, twelve BW-related QTLs were distributed across eight LGs [40], and in crucian carp, eight BW-related QTLs spanned five LGs [39]. Among these 17 growth QTLs, 11 growth QTLs were distributed on LG06, LG09, and LG17. Notably, two overlapping intervals were identified: BW and BWI on LG09 (44.543 cM–45.835 cM), and BW and BL on LG17 (13.520 cM-14.832 cM). These findings indicate that, although growth traits in *C. fuscus* are regulated by multiple genes, the regulatory genes are mapped to a few major chromosomes. Similar observations have been made in black carp [1] and pikeperch (*Sander lucioperca*) [46].

### 4.2. Transcriptome Enrichment Pathways Associated with Body Size in C. fuscus

To further explore the genes associated with the growth of *C. fuscus*, a comprehensive transcriptome analysis was conducted, identifying 3824 DEGs. Among these, 2252 genes were highly expressed, while 1572 had low expression. The functional roles and biological pathways of these DEGs were examined using GO and KEGG enrichment analyses.

In the GO enrichment analysis, the majority of DEGs were found to be enriched in biological processes (94 out of 110 pathways, 85%), a trend observed in other species as well, such as grass carp (*Ctenopharyngodon idella*) (42%), black porgy (52%), and Zig-Zag eel (*Mastacembelus armatus*) (60%) [47,48,49]. These results suggest that DEGs related to biological processes are likely central to the growth rate differences observed among individuals. The most significantly enriched pathways were related to organic metabolism, particularly “pyruvate metabolism” and “ATP metabolism”, consistent with the known biological roles of these genes. Pathways related to ATP precursor synthesis, including “ribonucleoside triphosphate biosynthetic process” and “ATP biosynthesis”, were predominantly enriched in small-sized individuals. In contrast, big-sized individuals exhibited greater enrichment in metabolic pathways for the synthesis of various organic compounds, such as “pyruvate metabolic process” and “regulation of cellular component biosynthesis”. The upregulation of the tricarboxylic acid cycle (TCA cycle) in bigger *C. fuscus* suggests an enhanced capacity to utilize TCA intermediates for biosynthetic pathways, thus supporting cellular function and homeostasis [50]. Conversely, smaller individuals relied on oxidative phosphorylation to rapidly generate large amounts of ATP, which is crucial for cell proliferation and tissue growth.

KEGG analysis of carbohydrate metabolism pathways revealed that genes related to glycolysis, the TCA cycle, and propionate metabolism were upregulated in bigger individuals, supporting the maintenance of cellular functions and homeostasis [51]. In contrast, smaller individuals showed a greater tendency to convert odd-chain fatty acids and amino acids into TCA cycle intermediates. This observation, consistent with previous studies [52], was supported by the increased expression of key enzymes involved in the catabolism of isoleucine and the β-oxidation of odd-chain fatty acids. These enzymes include acyl-coA oxidase 1 (*acox1*) [53], the lipoamide acyltransferase component of branched-chain alpha-keto acid dehydrogenase (*dbt*) [54], and ethylmalonyl-CoA decarboxylase (*echdc1*) [55], which facilitate the conversion of succinyl-CoA to succinate, releasing GTP [56,57]. This observation highlights the need for rapid energy generation in smaller individuals to support growth and proliferation.

### 4.3. Identification and Characterization of Growth-Associated Genes in C. fuscus

In this study, we conducted a comprehensive analysis integrating QTL mapping and transcriptome data to identify growth-associated candidate genes in *C. fuscus*. A total of 27 candidate genes were successfully identified across 17 QTL regions linked to growth traits. Notably, compared to small-sized individuals, several genes including *eya4*, *serca1*, *f11r*, *npm1*, and *tecta* were significantly upregulated in big-sized individuals. The *eya4* gene, part of the EYA gene family, has been implicated in various biological processes such as organogenesis and cell differentiation [58]. Previous studies have shown that *eya4* can influence the development of the zebrafish sensory system by modulating Na+/K ATPase (*atp1b2b*) activity [59]. Additionally, *eya4* has been found to dephosphorylate polo-like kinase 1 (*plk1*), thereby promoting mitotic progression and enhancing cell differentiation [60]. Similarly, *serca1*, a key subtype in the SERCA family, is primarily expressed in fast-twitch skeletal muscle and plays a crucial role in transporting Ca^2+^ from the cytoplasm to the sarcoplasmic reticulum (SR) cavity following muscle contraction, thereby promoting muscle relaxation. Recent studies have shown that serca1 is regulated by Sarcolipin (*sln*), with this regulation being particularly significant in fast-twitch skeletal muscle [61]. Additionally, *serca1* is involved in regulating intracellular calcium ion (Ca^2^⁺) concentrations during zebrafish embryonic development, influencing left–right asymmetry during embryogenesis. However, the specific downstream effector molecules of *serca1* remain unclear, and it is hypothesized that calmodulin (*cam*) or calmodulin-dependent protein kinase II (*camkii*) may be involved [62,63]. Other genes such as *f11r* [64], *npm1* [65], and *tecta* [66] have also been linked to cellular differentiation and organ development. The upregulation of these genes in big-sized individuals suggests their pivotal role in the growth and development of *C. fuscus*.

In contrast, small-sized individuals exhibited higher expression levels of genes such as *mstn*, *egfr*, *mto1*, *ikzf2*, and *mks1*. *Mstn*, a member of the transforming growth factor-β (TGF-β) superfamily, inhibits muscle growth by targeting myogenic regulatory factors such as *myog*, *myod*, and *myf5* [67,68]. Additionally, it can influence vertebrate growth by reducing the activity of cyclin-dependent kinase 2 (*cdk2*) and upregulating p21, a *cdk* inhibitor, thereby arresting myoblasts in the G1 phase of the cell cycle [69]. Gene knockout studies in comparative olive fish have shown that the disruption of *mstn* expression can stimulate myoblast proliferation and increase muscle mass, leading to greater fish weight [70,71]. Additionally, *egfr* plays a critical role during early development in species such as Japanese medaka (*Oryzias latipes*), where it is involved in cleavage, protocol formation, and organogenesis [72]. Other studies have shown that *egfr* is also associated with the maturation of zebrafish ovaries. In ovarian tissue, the *pi3k/akt* pathway mediates the regulation of *igf-I* on *egfr*, enhancing *egfr* expression and phosphorylation. This inhibition of receptor-type tyrosine protein phosphatase kappa (*ptprk*) activates the *egf/egfr* signaling pathway, promoting oocyte maturation [73]. The genes *mto1* [74], *ikzf2* [75], and *mks1* [76] are associated with embryonic development and organogenesis.

Previous studies indicate that these genes play key roles in embryonic development, cell differentiation, and proliferation, consistent with enrichment analyses showing a strong association with biological process-related pathways. Notably, a significant distinction was observed: genes upregulated in larger individuals predominantly supported sensory system development, while those upregulated in smaller individuals favored the development of various tissues and organs. This suggests that energy allocation differences, driven by divergent cellular differentiation processes, may underlie the growth and developmental variations observed between *C. fuscus* individuals [77]. However, further molecular validation, such as gene knockout or knockdown approaches, is needed to clarify the functional roles of candidate genes within the QTL intervals linked to growth traits in *C. fuscus*. QTL mapping will also be essential for MAS [78]. SNP molecular markers identified in candidate genes will be statistically analyzed and prioritized for population validation. By selecting markers with significant effects on growth and optimizing marker combinations, we aim to enhance the growth performance of *C. fuscus* across diverse environments.

## 5. Conclusions

In this study, we conducted QTL mapping for eight growth traits in *C. fuscus*, identifying 17 growth-related QTLs across eight intervals, which encompassed 162 functional genes. RNA-seq analysis of individuals with different sizes revealed 3824 differentially expressed genes, with the big-sized group showing 2252 highly expressed and 1572 low-expressed genes compared to the small-sized group. By integrating QTL mapping and RNA-seq data, we identified 27 candidate genes, including *eya4*, *serca1*, *f11r*, and *npm1*. This combined approach offered a more precise list of candidate genes for growth traits in *C. fuscus*, providing valuable genetic insights to enhance growth performance in this species.

## Figures and Tables

**Figure 1 animals-15-01707-f001:**
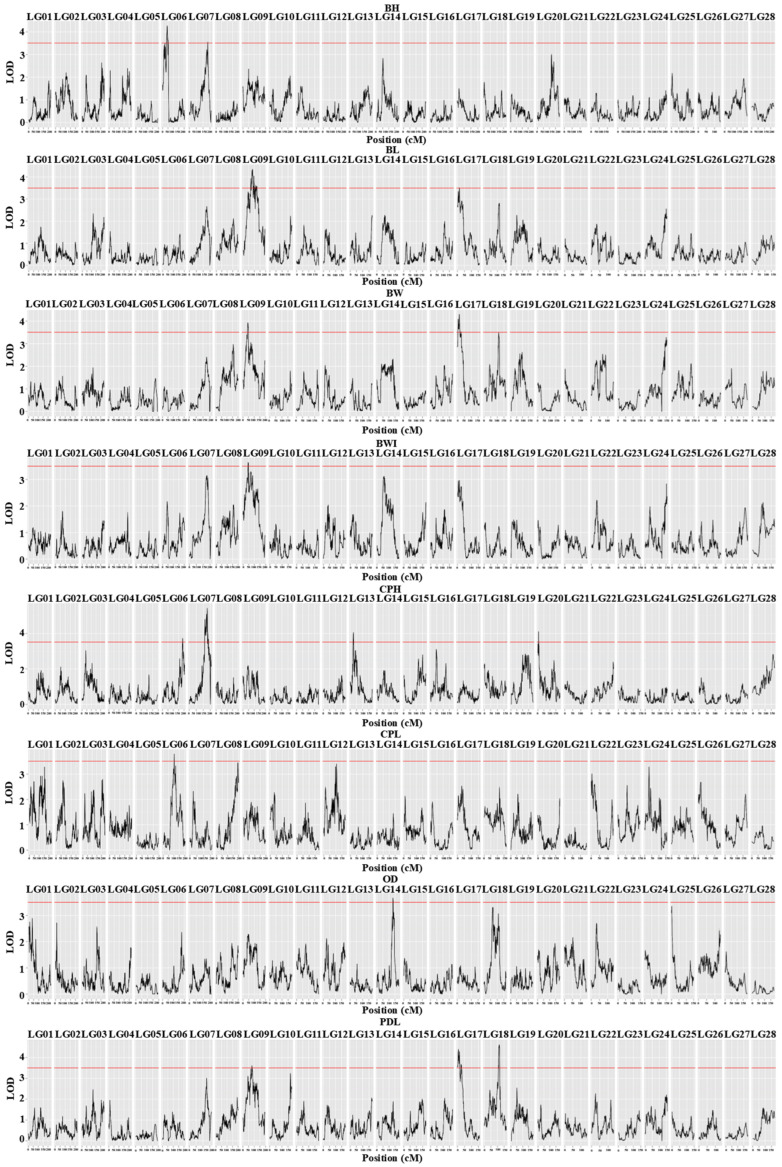
LOD scores along the 28 linkage groups for the variations in eight growth-related traits in *C. fuscus*. The horizontal axis represents the positions of the linkage groups, while the vertical axis shows the corresponding LOD values. The red dashed horizontal line indicates the LOD significance threshold of 3.5 for the linkage population.

**Figure 2 animals-15-01707-f002:**
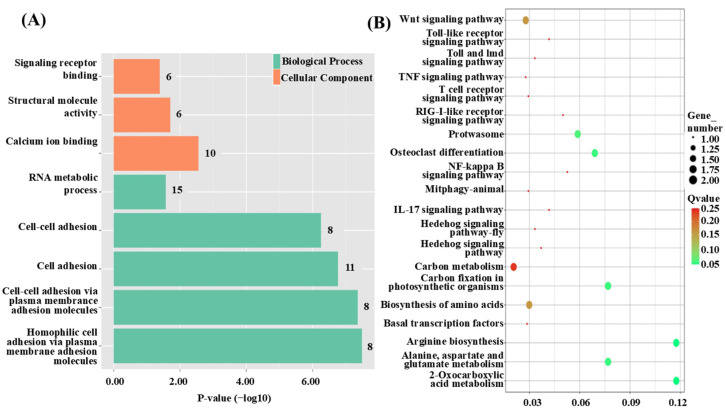
Functional enrichment analysis of QTL intervals for growth traits in *C. fuscus*. (**A**) Cluster diagram of GO enrichment analysis for QTL intervals associated with growth traits. The color of the bars indicates the gene enrichment category, while the bar length represents the significance level of gene expression. (**B**) Bubble plot of KEGG enrichment analysis for QTL intervals related to growth traits. The size of the circles reflects the number of genes, and the color of the circles indicates the significance level of gene expression.

**Figure 3 animals-15-01707-f003:**
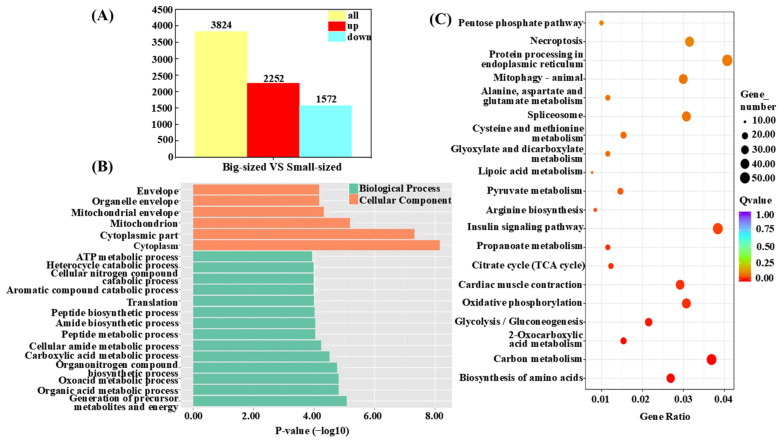
Transcriptional profiles of big-sized and small-sized groups for growth traits in *C. fuscus*. (**A**) Differentially expressed genes between big-sized and small-sized groups. (**B**) Analysis of the top 20 enriched GO terms in “Biological Processes” and “Cellular Components” based on adjusted *p*-values. The color of the bars indicates the gene enrichment category, and the bar length represents the significance level of gene expression. (**C**) KEGG pathway enrichment analysis of the top 20 adjusted *p*-values. The size of the circles represents the number of genes, and the color indicates the significance level of gene expression.

**Figure 4 animals-15-01707-f004:**
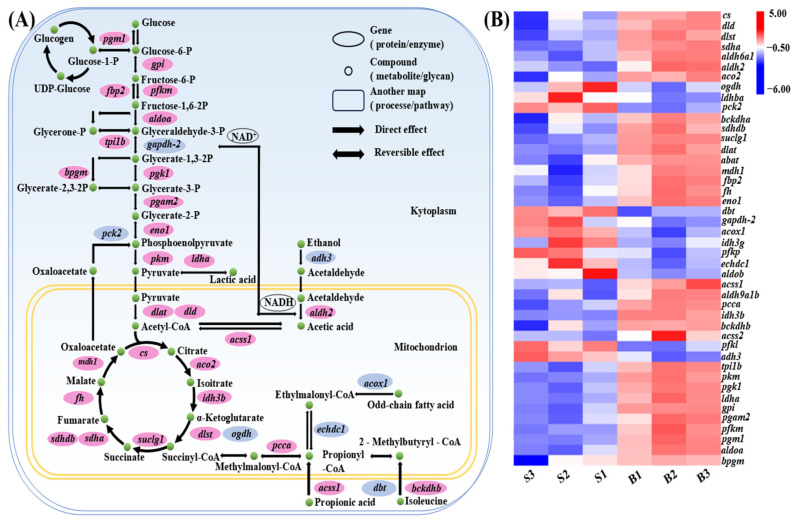
(**A**) Carbohydrate metabolism pathways in muscle tissue of *C. fuscus* (partial). Red indicates upregulated gene expression in big-sized individuals, while blue indicates upregulated gene expression in small-sized individuals. (**B**) Hierarchical clustering analysis of differentially expressed genes (DEGs) in carbohydrate metabolism pathways between big-sized and small-sized groups.

**Figure 5 animals-15-01707-f005:**
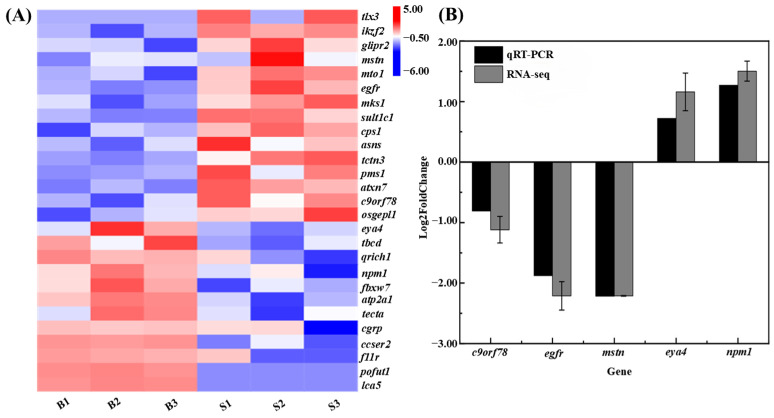
(**A**) Hierarchical clustering analysis of growth candidate genes jointly analyzed by QTL and RNA-seq. Red indicates upregulated gene expression, while blue indicates downregulated gene expression. (**B**) Comparative analysis of qRT-PCR results and RNA-seq data.

**Table 1 animals-15-01707-t001:** QTL mapping results for growth traits in *C. fuscus* using the composite interval mapping method.

Trait	QTL	Linkage Group	Marking Interval (cM)	Peak Marker (cM)	LOD	PVE (%)	LOD Peak (cM)	Corresponding Markers for LOD Peaks	Candidate Gene
BW	qBW-1	LG09	41.931–45.835	np27	3.910	8.70	44.543	LG9 6545823	
	qBW-2	LG17	13.52–17.098	np1240	4.280	9.50	14.832	LG17 28458728	*slc4a1*
BL	qBL-3	LG09	82.998–85.093	lm55	4.350	9.80	83.615	LG9 13097329	
	qBL-4	LG17	11.779–14.832	lm1144	3.510	8.00	13.520	LG17 28458720	*slc4a1*
CPH	qCPH-5	LG06	184.207–186.967	lm3244	3.730	8.30	186.516	LG6 4577776	*pgap1*
	qCPH-6	LG07	128.571–129.622	lm3434	5.440	11.90	128.590	LG7 8175232	
	qCPH-7	LG13	24.138–31.262	np671	4.060	9.00	26.118	LG13 30699773	*ltk*
	qCPH-8	LG20	5.037–6.976	lm1639	4.110	9.10	5.619	LG20 1586316	
CPL	qCPL-9	LG06	102.507–103.569	np3359	3.790	9.50	102.644	LG6 22085749	*pdk1*
OD	qOD-10	LG14	132.742–134.846	hk257	3.580	8.00	134.365	LG14 25875874	*qrich1*
PDL	qPDL-11	LG09	77.872–80.447	lm52	3.610	8.10	79.160	LG9 12492904	*pp2a*
	qPDL-12	LG17	2.533–6.695	np1243	4.400	9.80	5.627	LG17 29429338	*mto1*
	qPDL-13	LG18	112.055–115.500	lm1230	4.580	10.20	112.554	LG18 25096181	
BWI	qBWI-14	LG09	44.543–46.478	np28	3.630	8.10	45.835	LG9 6720412	
BH	qBH-15	LG06	21.046–22.095	np3410	3.500	8.00	21.431	LG6 36044360	*cps*
	qBH-16	LG06	41.649–44.768	np3396	4.220	9.60	41.702	LG6 34731920	
	qBH-17	LG07	134.41–134.773	lm3428	3.550	8.10	134.410	LG7 7360877	

**Table 2 animals-15-01707-t002:** Summary of transcriptome data generated from *C. fuscus* samples.

Sample	Raw Reads	Clean Reads	Clean Read Rate (%)	Total Map	Total Map Rate (%)	Q20	Q30
S1	49,640,720	47,460,188	95.61	41,564,096	87.58	98.16	95.10
S2	46,547,234	44,737,262	96.11	35,866,634	80.17	98.12	95.23
S3	42,787,644	40,724,206	95.18	33,787,631	82.97	98.01	94.76
B1	51,865,132	50,973,770	98.38	46,109,511	90.46	98.61	95.99
B2	50,052,990	49,099,298	98.09	43,891,232	89.39	98.76	96.40
B3	52,370,048	51,467,180	98.28	47,125,928	91.57	98.71	96.22

**Table 3 animals-15-01707-t003:** The top 20 genes in the transcriptome associated with body size in *C. fuscus*.

Gene ID	Gene Name	Gene Annotation	log_2_ (FC)
cfu_9G0006470	*trim32*	E3 ubiquitin-protein ligase TRIM32	9.534
cfu_15G0008560	*bloc-1*	Biogenesis of lysosome-related organelles complex 1	9.487
cfu_16G0005300	*wwtr1*	WW domain-containing transcription regulator protein 1	8.941
cfu_26G0005640	*scfd2*	Sec1 family domain-containing protein 2	8.833
cfu_1G0009900	*nipsnap3a*	Protein NipSnap homolog 3A	8.720
cfu_27G0004410	*camlg*	Guided entry of tail-anchored protein factor	8.636
cfu_10G0006000	*slc16a4*	Monocarboxylate transporter 5	8.621
cfu_26G0004610	*atraid*	All-trans retinoic acid-induced differentiation factor	8.559
cfu_27G0003260	*efemp2*	EGF-containing fibulin-like extracellular matrix protein 2	8.472
cfu_5G0009050	*aadac*	Arylacetamide deacetylase	8.306
cfu_3G0006820	*rims4*	Regulating synaptic membrane exocytosis protein 4	−6.820
cfu_9G0008290	*shc3*	SHC-transforming protein 3	−6.933
cfu_6G0004030	*lpa1*	High-affinity lysophosphatidic acid receptor	−6.988
cfu_15G0007480	*nos2*	Nitric oxide synthase	−7.002
cfu_3G0009150	*tcf23*	Transcription factor 23	−7.015
cfu_14G0005780	*hrh1*	Histamine H1 receptor	−7.032
cfu_4G0003010	*apoc-1*	Apolipoprotein C-I	−7.104
cfu_14G0006580	*chia*	Acidic mammalian chitinase	−7.329
cfu_11G0001850	*fgfbp2*	Fibroblast growth factor-binding protein 2	−7.474
cfu_22G0000210	*itln*	Intelectin	−7.841

**Table 4 animals-15-01707-t004:** Candidate genes associated with growth in *C. fuscus* identified based on QTL mapping and transcriptome analysis.

Gene ID	Gene Name	Gene Annotation	log_2_ (FC)
cfu_6G0008310	*calcrla*	calcitonin gene-related peptide type 1 receptor isoform X1	2.397
cfu_6G0008340	*wdr75*	WD repeat-containing protein 75	1.554
cfu_6G0008350	*asnsd1*	asparagine synthetase domain-containing protein 1	−1.380
cfu_6G0008380	*osgepl1*	probable tRNA N6-adenosine threonylcarbamoyltransferase	−1.102
cfu_6G0008410	*pms1*	PMS1 protein homolog 1	−1.197
cfu_6G0008420	*mstnb*	myostatin	−2.353
cfu_6G0008440	*pofut2*	GDP-fucose protein O-fucosyltransferase 2	6.582
cfu_6G0008510	*ikzf2*	IKAROS Family Zinc Finger 2	−2.744
cfu_6G0008520	*egfr*	epidermal growth factor receptor	−2.210
cfu_6G0008530	*cps1*	carbamoyl-phosphate synthase	−1.581
cfu_9G0003920	*c9orf78*	telomere length and silencing protein 1 homolog	−1.118
cfu_9G0003990	*tecta*	alpha-tectorin	2.098
cfu_9G0004000	*tbcel*	tubulin-specific chaperone cofactor E-like protein	1.174
cfu_9G0004050	*f11r*	junctional adhesion molecule A	4.879
cfu_9G0003810	*glipr2*	Golgi-associated plant pathogenesis-related protein 1	−2.428
cfu_9G0002330	*mks1*	Meckel syndrome type 1 protein	−1.804
cfu_14G0007230	*qrich1*	glutamine-rich protein 1	1.404
cfu_17G0007930	*atxn7l3*	ataxin-7protein 3	−1.171
cfu_17G0008220	*mto1*	protein MTO1 homolog	−2.294
cfu_17G0008240	*tctn3*	tectonic-3	−1.298
cfu_17G0008260	*sult1c1*	sulfotransferase family cytosolic 1B member 1	−1.707
cfu_17G0008270	*atp2a1*	sarcoplasmic/endoplasmic reticulum calcium ATPase 1	1.895
cfu_17G0008530	*ccser2*	serine-rich coiled-coil domain-containing protein 2	2.420
cfu_18G0007570	*lca5*	lebercilin	7.172
cfu_18G0007610	*eya4*	eyes absent homolog 4 isoform X2	1.161
cfu_20G0000670	*tlx3*	T-cell leukemia homeobox protein 3	−6.172
cfu_20G0000680	*npm1*	nucleophosmin	1.503

## Data Availability

RNA-Seq data have been deposited in the NCBI database under the project accession number PRJNA1258690.

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
