# Peer review of "Integrative QTL Mapping and Transcriptomic Profiling to Identify Growth-Associated QTL and Candidate Genes in Hong Kong Catfish (Clarias fuscus)"

_animals, 2025, doi:10.3390/ani15121707_

Round 1
Reviewer 1 Report
Comments and Suggestions for Authors
This study conducted a systematic analysis of the genetic regulatory mechanisms influencing growth traits in Hong Kong catfish (Clarias fuscus) by integrating quantitative trait locus (QTL) mapping with transcriptome analysis. A genetic map was developed using data from 200 full-sibling individuals, which led to the identification of 17 QTL intervals associated with eight growth traits, including body weight, body length, and body height, with a phenotypic variance explained (PVE) ranging from 8.00% to 11.90%. Additionally, by performing a differential transcriptome analysis on muscle tissues from groups of large and small catfish, 3,824 differentially expressed genes (DEGs) were identified. Employing a co-localization strategy, 27 core candidate genes were pinpointed, among which were myostatin (mstnb), epidermal growth factor receptor (egfr), and the calcium pump gene (serca1). These genes are believed to play significant roles in metabolism, myocyte proliferation, and embryonic development. The insights gained from this research offer valuable targets for the molecular breeding of Hong Kong catfish.
The paper presents several issues that need to be addressed for clarity and depth. Firstly, the PVE values of the QTL intervals are notably low, with all values being less than 12%. The authors have not discussed the implications of these low polygenic additive effects, which is a critical aspect that should be elaborated upon in the discussion section. Secondly, while the authors have provided detailed annotations regarding the functions of candidate genes, including the novel role of EGFR in embryonic development, they have not validated these candidate genes through experimental methods such as CRISPR knockout or overexpression. This lack of validation makes the inference of functional mechanisms appear largely theoretical. It is advisable for the authors to discuss the target genes more thoroughly in the discussion and consider designing experiments to validate their findings. Lastly, although the research findings have the potential to guide molecular marker-assisted breeding aimed at enhancing the growth performance of Hong Kong catfish, the paper fails to propose a concrete breeding strategy, such as a marker combination selection strategy or a pathway for industrialization. The description of potential applications remains vague and would benefit from more specific recommendations.
Reviewer 2 Report
Comments and Suggestions for Authors
REVIEW OF THE PAPER TITLED ‘INTEGRATIVE QTL MAPPING AND TRANSCRIPTOMIC PROFILING IDENTIFY GROWTH-ASSOCIATED QTL AND CANDIDATE GENES IN HONG KONG CATFISH (CLARIAS FUSCUS)’
S/NO |
SECTION/ LINES/PAGES |
COMMENTS |
1. 1 |
Abstract |
The abstract, being a standalone section, does not clearly state the knowledge gap addressed by the authors. Please include a statement in line 17 to explicitly highlight this gap. |
2. 2 |
Introduction
|
The study objective presented in Lines 113–114—"However, the genetic mechanisms underlying growth traits remain poorly understood,"—along with the aim stated in the abstract (line 27), which refers to identifying candidate genes associated with growth in C. fuscus through the integration of QTL mapping and RNA-seq analysis of DEGs between two extreme body size groups, is currently unclear and somewhat confusing. The authors are advised to revise and harmonize these statements to present a clear, consistent study objective throughout the manuscript. |
3. |
Materials and Methods |
The authors need to address the following points to enhance the clarity and rigor of the methodology:
|
4. |
Line 138 |
Which version of SPSS software was used |
5. |
Line 143 |
The authors state that “At 6 months of age, 254 individuals were randomly selected for body weight measurement,” which appears inconsistent with the figure of 200 subjects mentioned in the abstract (line 33). This discrepancy raises an important question: How many catfish were used in the study? The authors need to clarify the total number of subjects involved and reconcile this inconsistency to ensure accuracy and transparency in the study’s design and reporting. |
6. |
Line 144 |
Line 144 states that “from these, the 9 largest and 9 smallest individuals were chosen to form 3 big-sized groups (B1, B2, B3) and 3 small-sized groups (S1, S2, S3),” but the basis for this classification is unclear. The authors should clarify how the largest and smallest individuals were determined—for example, was it purely based on body weight, length, or another growth metric? Additionally, they should specify whether any statistical thresholds or percentile cut-offs were applied. This clarification is necessary to ensure that the grouping criteria are transparent and scientifically sound. |
7. |
Line 172 |
The authors should briefly indicate the RNA extraction method used in the study. For example, if a specific kit or protocol (such as TRIzol reagent, column-based kits, or a phenol-chloroform method) was employed, it should be clearly stated, including any modifications made to the standard protocol. This information is essential for reproducibility and assessment of RNA quality and integrity. |
8. |
Transcriptome Library Creation and Raw Data Processing |
The manuscript does not provide details on how sequencing was performed. The authors should clearly state:
Including these details is crucial for ensuring reproducibility and assessing the reliability of the RNA-seq data.
|
9. |
RESULTS |
The authors should consider moving Figure I on page 6 to the supplementary material, as it primarily presents quality check results rather than a core output of the study. While important for validating the data used in subsequent analyses, it does not directly contribute to the main findings and would be more appropriately placed as supporting information. |
10. |
All Figures |
The authors are advised to use high-resolution versions of all figures, particularly Figures 2, 3, and 4, to ensure clarity and readability. The current resolution may hinder the reader's ability to interpret critical details, such as gene labels, axis titles, and data points. Enhancing figure quality will significantly improve the presentation and overall comprehension of the study's results. |
|
Discussion |
|
11. |
Lines 341-349 |
The authors have not adequately cited previous related studies. To strengthen the scientific context and demonstrate how their work builds upon or differs from existing research, they should enhance the literature review by incorporating and discussing relevant similar studies. This includes citing previous work on growth trait genetics, QTL mapping, and RNA-seq analysis in C. fuscus or other closely related fish species. Doing so will not only provide a more comprehensive background but also highlight the novelty and significance of the current investigation.
|
12. |
Lines 439-441 |
The statement, “Our findings suggest that these genes generally promote embryonic development, cell differentiation, and proliferation, consistent with observations from enrichment anal- which showed that many genes are enriched in biological process-related pathways,” may imply that the results were not experimentally validated. To maintain scientific transparency and credibility, the authors should explicitly acknowledge this as a limitation of the study. Clearly stating that the findings are based on bioinformatics predictions without functional validation will help set appropriate expectations and guide future research directions.
|
Round 2
Reviewer 1 Report
Comments and Suggestions for Authors
The authors have made revisions in accordance with my review comments and have met the requirements of the journal for publication and are recommended for publication.
Reviewer 2 Report
Comments and Suggestions for Authors
NA